# Topological orders in (4+1)-dimensions

**Theo Johnson-Freyd[1,2] and Matthew Yu[1⋆]**

**1** Perimeter Institute for Theoretical Physics, Waterloo, Ontario
**2** Department of Mathematics, Dalhousie University, Halifax, Nova Scotia

⋆ myu@perimeterinstitute.ca

## Abstract

We investigate the Morita equivalences of (4+1)-dimensional topological orders. We show that any (4+1)-dimensional super (fermionic) topological order admits a gapped boundary condition — in other words, all (4+1)-dimensional super topological orders are Morita trivial. As a result, there are no inherently gapless super (3+1)-dimensional theories. On the other hand, we show that there are infinitely many algebraically Morita-inequivalent bosonic (4+1)-dimensional topological orders.

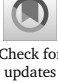

## 1 Introduction

A physical system described by a Hamiltonian is *gapped* when the spectrum of eigenvalues for the Hamiltonian has a gap between the lowest energy state and the vacuum. Such systems

prevent the existence of particles that are arbitrarily light. A *gapped phase* is an equivalence class of gapped systems. Systems that can be continuously deformed into each other without closing the energy gap are considered to be in the same phase. The low-energy limit of gapped phases may exhibit topological behaviour. Such is true for some quantum field theories, which flow in the infrared to topological theories [1]. All of the dynamical degrees of freedom can be integrated out, leaving only the topological excitations. The study of gapped phases in various dimensions has led to interest regarding the topological nature of extended objects, or operators, in these phases. In nontrivial cases, the content of operators and defects, as well as the algebraic structure of how they interact, compile into a topological order [2].

The classification of topological orders has been an interesting problem that combines the mathematics of higher category theory with the physics of gapped topological phases. By now the classification in lower dimensions is understood. In (1+1)-dimensions, topological orders are classified by their spectrum of point operators together with anomaly information that manifests as a class in ordinary or supercohomology. Some other well studied situation are in (2+1)d where topological orders with nondegenerate local ground states are classified by modular tensor categories [3], and in (3+1)d where topological order with nondegenerate local ground states are (modulo a few subtleties) always described by finite group gauge theories [4–7].

This paper addresses the classification in (4+1)d. We focus on the case of super topological orders, i.e. topological orders defined over the category **SVec** of super vector spaces, because the existence of super fibre functors makes this case technically easier. Following the strategy of [4,5], the first step is to condense out all of the line operators in the topological order. The resulting topological order has no line operators, and our first result is a classification of these:

$$\{\text{super (4+1)d topological orders with no lines}\} = \{\text{symplectic finite Abelian groups}^{1}\}. \quad (1)$$

By reducing along a Lagrangian subgroup, we furthermore show that every super (4+1)d topological order can be condensed all the way to the vacuum via a gapped topological boundary:

$$\{\text{super (4+1)d topological orders}\}/\text{Morita equivalence}^{2} = \{1\}. \quad (2)$$

This is to be expected, as it agrees with the cobordism classification proposed by [8]: a Morita-nontrivial super (4+1)d topological order should have a nontrivial gravitational anomaly detectable on (5+1)d spin manifolds, but every (5+1)d spin manifold is spin-nullcobordant.

By studying a spectral sequence introduced in [6,7], the classification (2) allows us to compute the analogous group for bosonic topological orders. We find that there is an isomorphism:

$$\{\text{bosonic (4+1)d topological orders}\}/\text{Morita equivalence} \cong \mathbb{Z}_2^{\infty}. \quad (3)$$

In other words, there are infinitely many pairwise-Morita-inequivalent bosonic (4+1)d topological orders (and each has a gapped boundary to its time-reversal). This disagrees with the cobordism prediction: the cobordism group of (5+1)d oriented manifolds is trivial. The origin of the disagreement, and indeed of the answer (3), is in (2+1)d: the Witt groups $\mathcal{W}$ and $\mathcal{SW}$ of Morita equivalence classes of bosonic and super modular tensor categories, studied in [9], are very large, whereas the cobordism classification would have predicted a classification in terms of the central charge alone.

The outline of our paper is as follows. Section 2 starts off by explaining how to reduce the set of operators in a (4+1)d topological order to only the surface operators, and how to see that

---

[1]We give the definition of symplectic finite Abelian group at the start of section 3.

[2]By definition, two topological orders are Morita equivalent if they be separated by a gapped topological interface.

their monoidality is given by a finite Abelian group. In principle this procedure works for the bosonic and fermionic case, up to a small caveat that is remarked upon. In that section though, we give the explanation specifically for super topological orders. We then review some aspects of fusion and sylleptic 2-categories to understand the nature of how surface operators pair up given three ambient dimensions. The build up is to see by way of a cohomology calculation that (4+1)d topological orders are parametrized by a symplectic form carried by the finite group of surface operators, establishing (1).

Section 3 outlines the method of symplectic reduction and its relation to Morita equivalence. This allows us to prove (2) that (4+1)d super topological orders all admit a gapped boundary. We furthermore give relationships between the bulk and boundary theories, where we interpret the bulk (4+1)d theory to be a higher form of "centre" for the boundary theory. To juxtapose with Section 2, we present a *bosonic* example of how the centre construction goes through. Lastly, we address the question of lifting boundary theories into the bulk, and obstructions in doing so.

Section 4 explains how we recover a bosonic theory from a fermionic theory plus extra information in "descent data", and computes the group of Morita equivalence of (4+1)d bosonic topological orders.

In many parts of the paper we will also draw analogies to lower dimensional theories when instructive.

## 2 5-dimensional Super Topological Orders

### 2.1 Condensing out the lines

An $(n+1)$-dimensional super topological order is defined in [6, 10–12] to be a multifusion $n$-supercategory $\mathcal{A}$ with trivial centre.[3] Triviality of the center is an axiomatization of the principle of remote detectability. For our purposes we will be considering only the *fusion* case. By this, we mean that there are no nontrivial 0-dimensional operators. This is to say that the ground state of our topological order is nondegenerate [15]. The principle of remote detectability, along with the fusion condition, implies that all codimension-1 operators arise as condensation descendants [6, Theorem 4]. In an arbitrary 5d[4] topological order given by the fusion 4-category $\mathcal{A}$, we therefore only need to consider operators of codimension-2 and higher. We will focus on the *super* case in which $\mathcal{A}$ is enriched over **SVec**.

We will deem two 5d topological orders as being *Morita equivalent* if they can be separated by a gapped 4d topological interface; this is also known as *Witt equivalence*. One way to produce a Morita equivalence is to perform a categorical condensation [16], where the condensation wall that separates the two phases is gapped and described by its own higher category of operators.

The first main step in our classification of 5d topological orders is to use the method outlined in [4,5] to condense out all the lines in any super 5d topological order. Here is a streamlined version of their construction, written in the language of [6,16]:

Within the super fusion 4-category $\mathcal{A}$ describing the topological order, there is a symmetric super fusion 1-category $\Omega^3 \mathcal{A}$ of line operators. Suppose that we choose a functor $F : \Omega^3 \mathcal{A} \to$ **SVec** of symmetric super fusion 1-categories. Such $F$ is called a *fibre functor*, and in the super case always exists [17]; since $\mathcal{A}$ is assumed to be fusion, $F$ is unique up to isomorphism, although not up to unique isomorphism.

---

[3]All of our "$n$-categories" are "weak." For example, a "2-category" is a bicategory. Multifusion 2-categories were first introduced by [13], and the n-category generalization was developed in [6,14].

[4]For the remainder of this paper whenever the dimension of an extended object or phase is given without the time component specified, we will take that dimension to represent the full spacetime dimension.

This $F$ can be "suspended" to a functor $\Sigma^3 F : \Sigma^3 \Omega^3 \mathcal{A} \to \Sigma^3 \mathbf{SVec}$, where $\Sigma^3 \Omega^3 \mathcal{A} \subset \mathcal{A}$ is the sub 4-category of operators which arise as condensation descendants from line operators, and $\Sigma^3 \mathbf{SVec}$ is the 4-category of operators in the vacuum 5d super topological order. This $\Sigma^3 F$ makes the 4-category $\Sigma^3 \mathbf{SVec}$ into a module for the fusion 4-category $\Sigma^3 \Omega^3 \mathcal{A}$. We may induce (aka base change) this module along the inclusion $\Sigma^3 \Omega^3 \mathcal{A} \subset \mathcal{A}$ to produce an $\mathcal{A}$-module

$$\mathcal{M} := \mathcal{A} \otimes_{\Sigma^3 \Omega^3 \mathcal{A}} \Sigma^3 \mathbf{SVec}.$$

We set $\mathcal{B} := \mathrm{End}_{\mathcal{A}}(\mathcal{M})$ to be the super fusion 4-category of $\mathcal{A}$-linear endomorphisms of $\mathcal{M}$; then $\mathcal{M}$ is a Morita equivalence $\mathcal{A} \simeq \mathcal{B}$[5]. Because we started with a fibre functor on the full category of line operators in $\mathcal{A}$, there are no nontrivial line operators in $\mathcal{B}$, i.e. $\Omega^3 \mathcal{B} = \mathbf{SVec}$.

**Remark.** In the case of bosonic topological orders, to condense all the lines would require choosing a bosonic fibre functor $\Omega^3 \mathcal{A} \to \mathbf{Vec}$. Such a functor exists if and only if there are no emergent fermions [17].

Since $\Omega^3 \mathcal{B} = \mathbf{SVec}$, the result of [6, Theorem 5] implies that $\mathcal{B} = \Sigma^2 \mathcal{C}$, where $\mathcal{C} := \Omega^2 \mathcal{B}$ is the (sylleptic) fusion 2-category of surface operators (and junctions between them); the statement $\mathcal{B} = \Sigma^2 \Omega^2 \mathcal{B}$ means that all three- and four-dimensional "membrane" objects can be built as condensation descendants of surface operators. But $\Omega \mathcal{C} = \Omega^3 \mathcal{B} = \mathbf{SVec}$, i.e. it is *strongly super fusion*:

**Definition 2.1** ( [18]). *A super fusion 2-category $\mathcal{C}$ is* strongly super fusion *if* $\Omega \mathcal{C} := \mathrm{End}_{\mathcal{C}}(1) \cong \mathbf{SVec}$.

An object in a (super) fusion 2-category is *indecomposable* if it is nonzero and cannot be written as a direct sum of nonzero objects; recall from [13] that in a (super) fusion 2-category, an object is indecomposible iff it is simple. Two indecomposable objects are in the same *component* if they are related by a nonzero morphism; the set of components of a (super) fusion 2-category $\mathcal{C}$ is denoted $\pi_0 \mathcal{C}$. The second main step in our classification of 5d topological orders is a classification of strongly fusion 2-categories that we established in [18]:

**Theorem 2.2** ( [18, Theorem B]). *If $\mathcal{C}$ is a (super) fusion 2-category with $\Omega \mathcal{C} \cong \mathbf{SVec}$, then every indecomposable object of $\mathcal{C}$ is invertible. The equivalence classes of indecomposable objects in $\mathcal{C}$ form a finite group, which is a central double cover of the group $\pi_0 \mathcal{C}$ of components of $\mathcal{C}$ (in particular, $\pi_0 \mathcal{C}$ is a group).*

Since an invertible object always has the same endomorphisms as the identity, Theorem 2.2 implies in particular that the endomorphisms of any indecomposable object in $\mathcal{C}$ is equivalent to $\mathbf{SVec}$, a super version of the condition called "endotriviality" in [13].

## 2.2 Sylleptic and symplectic groups: bosonic case

In any 5d topological order, the surface operators have three ambient dimensions in which they can compose. Thus the fusion 2-category $\mathcal{C}$ is 3-monoidal, aka *sylleptic*. The definition of sylleptic monoidal 2-category, which can be found in full in the appendix of [19], simplifies dramatically in the strongly fusion case.

To warm up, in this section we discuss the case of bosonic strongly fusion 2-categories, where sylleptic structures are classified by the Eilenberg–MacLane cohomology introduced in [20]. Indeed, suppose that $\mathcal{C}$ is bosonic strongly fusion, meaning that it is a fusion 2-category with $\Omega \mathcal{C} = \mathbf{Vec}$. The bosonic case of Theorem 2.2 is [18, Theorem A], which says that the indecomposable objects in $\mathcal{C}$ form a finite group $M$, equal to the group of components since $\mathcal{C}$ is forced to be endotrivial.

---

[5]This construction presently outlined also goes by the name *deequivariantization*.

The full data of the monoidal structure on $\mathcal{C}$ consists of: a tensor functor $\otimes$, given by the group law on $M$; an associator $\alpha_{x,y,z} : (x \otimes y) \otimes z \xrightarrow{\sim} x \otimes (y \otimes z)$; and a pentagonator $\pi_{x,y,z,w}$

$$
\begin{array}{c}
(w \otimes x) \otimes (y \otimes z) \\
\qquad \alpha \nearrow \qquad\qquad \alpha \searrow \\
((w \otimes x) \otimes y) \otimes z \qquad\qquad \Uparrow \pi \qquad\qquad w \otimes (x \otimes (y \otimes z)) \quad , \\
\alpha \otimes I \downarrow \qquad\qquad\qquad\qquad\qquad\qquad \uparrow I \otimes \alpha \\
(w \otimes (x \otimes y)) \otimes z \xrightarrow{\qquad\qquad \alpha \qquad\qquad} w \otimes ((x \otimes y) \otimes z)
\end{array}
\tag{4}
$$

which must satisfy a certain equation that we will not reproduce in full. But by endotriviality, $\alpha$ is no data: there is up to isomorphism a unique equivalence $(x \otimes y) \otimes z \xrightarrow{\sim} x \otimes (y \otimes z)$ for every triple of indecomposable object $(x, y, z)$. After trivializing $\alpha$, the equation for $\pi$ says simply that it is a 4-cocycle in ordinary group cohomology with coefficients in $\mathbb{C}^\times$. We will henceforth adopt the following notation. Given a group $M$ (Abelian if $n \geq 2$), we will write $M[n]$ for the Eilenberg–Mac Lane space more typically written $K(M, n)$, and $\mathrm{H}^k(-)$ without coefficients always means ordinary cohomology with $\mathbb{C}^\times$ coefficients $\mathrm{H}^k(-; \mathbb{C}^\times)$. To summarize the above discussion, we find that bosonic strongly fusion 2-categories with $\mathcal{C}$ with $\pi_0 \mathcal{C} = M$ are classified by

$$
[\pi] \in \mathrm{H}^4_{\mathrm{gp}}(M) := \mathrm{H}^4(M[1]; \mathbb{C}^\times). \tag{5}
$$

Suppose $\mathcal{C}$ is a monoidal 2-category with tensor bifunctor $\otimes$, associator $\alpha$, and pentagonator $\pi$. A *braiding* on $\mathcal{C}$ consists of a natural (in both variables) equivalence $b_{x|y} : x \otimes y \to y \otimes x$,[6] together with *hexagonators* $R_{(x|-,-)}$ and $S_{(-,-|x)}$ that provide the monoidality of $b$:

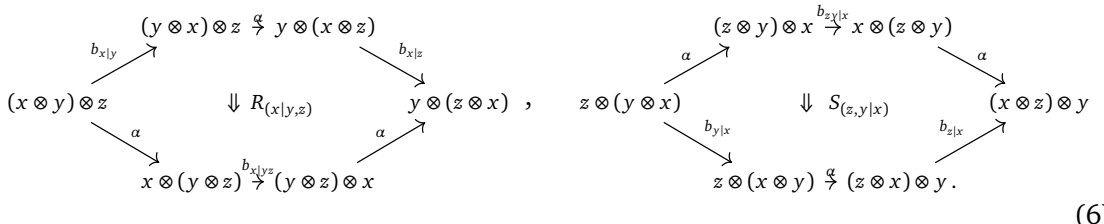

$$\tag{6}$$

$R$ and $S$ must solve various equations. When $\alpha$ and $\pi$ are trivial, these equations say first that for each $x$, $R_{(x|-,-)}$ and $S_{(-,-|x)}$ are 2-cocycles,[7] and they furthermore assert:

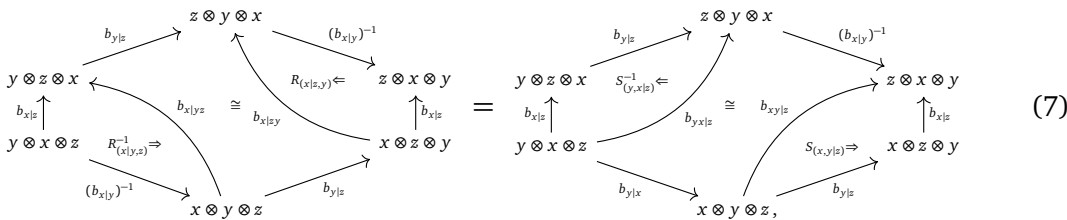

$$\tag{7}$$

---

[6] We write the braiding as $b_{x|y}$ rather than $b_{x,y}$ to be consistent with later notation for Eilenberg–Mac Lane cocycles. Higher Eilenberg–Mac Lane cocycles are like AT&T sales pitch: "More bars in more places."

[7] These are 3-cochains if we include the $x$ variable, but not ordinary 3-cocycles.

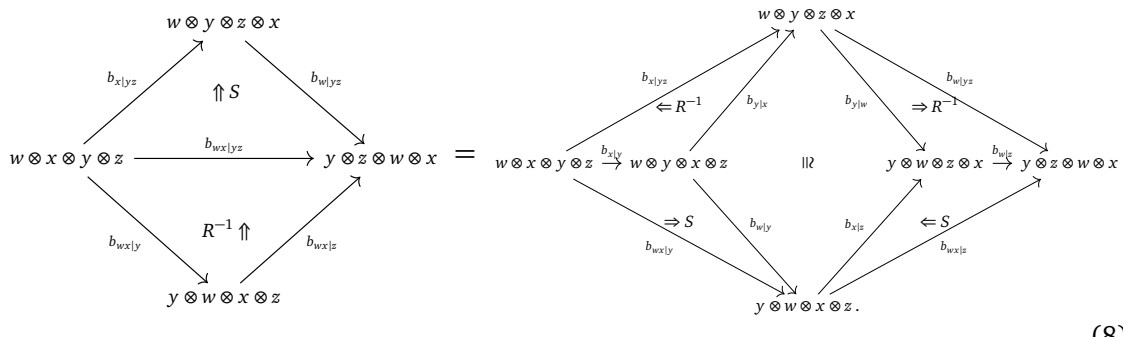

$$\text{(8)}$$

The unlabeled isomorphisms are the naturality of $b$. If $b$ is also trivial, then (7) and (8) simply say:

$$R^{-1}_{(x|y,z)} R_{(x|z,y)} = S^{-1}_{(y,x|z)} S_{(x,y|z)}, \tag{9}$$

$$R^{-1}_{(wx|y,z)} S_{(w,z|yz)} = R^{-1}_{(x|y,z)} R^{-1}_{(w|y,z)} S_{(w,x|y)} S_{(w,x|z)}. \tag{10}$$

Suppose that we are in the bosonic strongly fusion case. Then $\alpha$ and $b$ are automatically trivial, but $\pi$ may not be. In this case, the equivalent equations (8) and (10), as well as the requirements that $R_{x|-,-}$ and $S_{-,-|x}$ be 2-cocycles, receive corrections by $\pi$. (The equivalent equations (7) and (9) do not require corrections, because $\pi$ only appears when we need to coherently tensor four or more objects.) The full result is that $(\pi, R, S)$ are together the data of what is sometimes called an "Abelian cocycle," and what we will call a *braided cocycle*: they define a class in the Eilenberg–Mac Lane cohomology

$$[\pi, R, S] \in \mathrm{H}^4_{\mathrm{br}}(M; \mathbb{C}^\times) := \mathrm{H}^5(M[2]; \mathbb{C}^\times). \tag{11}$$

Finally, suppose that $\mathcal{C}$ is a braided monoidal 2-category. A *syllepsis $v$* for $\mathcal{C}$ is an isomorphism $v_{x\|y} : b_{x|y} \overset{\sim}{\Rightarrow} b^{-1}_{y|x}$ for each $x, y$ such that the diagram

$$
\begin{array}{ccc}
b_{\ell|xy} & \overset{R_{(\ell|x,y)}}{\Longleftarrow} & b_{\ell|x}\, b_{\ell|y} \\
\big\Downarrow v_{\ell,xy} & & \big\Downarrow v_{\ell\|x}\, v_{\ell\|y} \\
b^{-1}_{xy|\ell} & \underset{S_{(x,y|\ell)}}{\Longleftarrow} & b^{-1}_{x|\ell}\, b^{-1}_{y|\ell}
\end{array}
\tag{12}
$$

commutes. In the bosonic strongly fusion case where $\alpha$ and $b$ are trivial, $v$ enhances $(\pi, R, S)$ to a *sylleptic cocycle*, defining a class

$$[\pi, R, S, v] \in \mathrm{H}^4_{\mathrm{syl}}(M; \mathbb{C}^\times) := \mathrm{H}^6(M[3]; \mathbb{C}^\times). \tag{13}$$

In general, a theory with (only) grouplike $p$-spacetime dimensional objects with $q$-ambient dimensions (hence $p + q$ total spacetime dimensions) should be classified by degree $(p + q + 1)$ cohomology of $M[q]$[8]. The original paper [20] calculates the values of $\mathrm{H}^p(A[q]; B)$ for small

---

[8]When $p$ is large, the required cohomology theory is not ordinary cohomology. Indeed, any theory will have $k$-dimensional operators built by inserting decoupled $k$-dimensional topological theories, and for large enough $k$ there are nontrivial invertible $k$-dimensional topological field theories. For most purposes the presence of these decoupled operators does not affect the physics. However, these operators can arise as "higher fusion coefficients" for fusion of lower-dimensional operators. The result of this is that classifications by ordinary cohomology must be corrected in high dimensions.

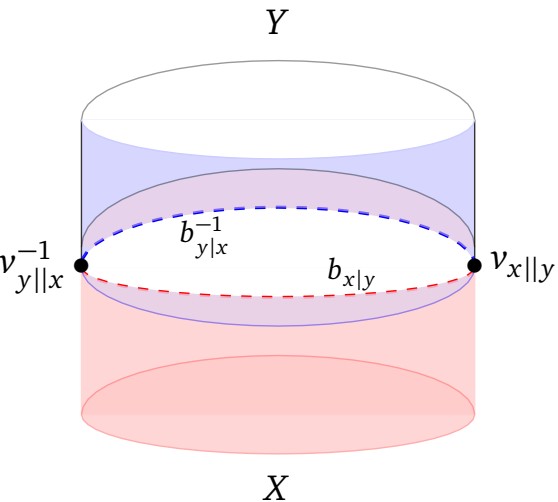

Figure 1: The two domed cylinders in red and blue represent two objects $X, Y \in \mathcal{C}$ respectively, living in four dimensions. The purple coloured regions show the domes of the objects. Initially, we can think of one object being above the other. The dashed lines indicate places where the two sheets pass over each other in the fourth dimension, with the colour indicating which is above. The two marked points show where one of the surfaces crosses over the other in the fifth dimension, changing the order of which surface is above and below. The change in color of the dotted circle represents the fact that after the syllepsis, the object which was initially on top, is now on the bottom.

values of $p, q$ and arbitrary Abelian groups $A, B$. In particular, writing $\widehat{A} := \hom(A, \mathbb{C}^{\times})$ and $M_2 := \hom(\mathbb{Z}_2, M)$, sylleptic strongly fusion 2-categories $\mathcal{C}$ with $\pi_0 \mathcal{C} = M$ are classified by

$$\mathrm{H}^6(M[3]; \mathrm{U}(1)) \cong \widehat{M_2} \oplus \widehat{\bigwedge^2 M},$$

where $\bigwedge^2 M := \frac{M \otimes M}{(m \otimes m)}$ denotes the alternating 2-forms on $M$. We will now explain the meaning of these two summands $\widehat{M_2}$ and $\widehat{\bigwedge^2 M}$. Further discussion can be found in [21, §2.1].

The summand $\widehat{M_2}$ measures the following [22]: given an invertible surface operator $m \in M$, consider wrapping the surface operator around a Klein bottle. This requires choosing an equivalence $m \cong m^{-1}$, since the Klein bottle is not orientable. We have such an equivalence exactly when $m \in M_2$, in which case, by endotriviality, the equivalence is unique up to isomorphism. It also requires choosing a Pin structure on the Klein bottle; let's choose the nonbounding Pin structure. Then this Klein bottle wrapped with $m \in M_2$ will evaluate to some element of $\mathbb{C}^{\times}$. This gives the map $M_2 \to \mathbb{C}^{\times}$, or in other words the element of $\widehat{M_2}$. Since the Klein bottle embeds into $\mathbb{R}^4 \subset \mathbb{R}^5$, this class in $\widehat{M_2}$ depends only on the braiding data and not the sylleptic form.

The summand $\widehat{\bigwedge^2 M}$ measures the following. Given surfaces with three ambient dimensions, then to "braid" them means passing them around each other in a two-parameter family, topologically a two-sphere. This procedure results in a phase factor that depends antisymmetrically on the inputs. In terms of the data of a sylleptic 2-category, this antisymmetric pairing is given by $\omega(x, y) = v_{x\|y} - v_{y\|x}$, where $v$ is a 2-cocycle and represents the sylleptic data. This is because $v$ tells how the surfaces go from above to below one another in the four dimension when we consider the double braiding of two surfaces. At two locations, the surfaces switch places by going into the fifth dimension. This process is depicted in Figure 1.

## 2.3 Sylleptic and symplectic groups: fermionic case

We turn now to the fermionic case, which is the main focus of this paper. As explained in §2.1, we are specifically interested in sylleptic strongly fusion super 2-categories $\mathcal{C}$. By definition, the line operators in such a 2-category are $\Omega\mathcal{C} = \mathbf{SVec}$. The simple lines consist of the identity line 1 and a fermion line $f$, corresponding to the super vector spaces $\mathbb{C}^{1|0}$ and $\mathbb{C}^{0|1}$ respectively. By Theorem 2.2, the components $\pi_0\mathcal{C}$ form a group $M$. The identity component, and hence every component, contains two simple objects. This identity component is a copy of $\Sigma\mathbf{SVec}$, equivalent to the 2-category of superalgebras and their super bimodules. The identity object $\mathbb{1}$ corresponds to the superalgebra $\mathbb{C}$, and the other simple object, which following [23] we will call the *Cheshire object* $c$, corresponds to the superalgebra Cliff(1). It is a fun exercise that the self-braiding $c \otimes c \to c \otimes c$ is given by the fermion $f$ [24]. The invertible operators in the identity component form the symmetric monoidal higher group $(\Sigma\mathbf{SVec})^\times = \mathbb{C}^\times[2].\{1,f\}[1].\{\mathbb{1},c\}[0]$ where the Postnikov extension data are given by $\mathrm{Sq}^2 : \{\mathbb{1},c\} \to \{1,f\}$ and $(-1)^{\mathrm{Sq}^2} : \{1,f\} \to \mathbb{C}^\times$.

The collection of invertible operators in $\mathcal{C}$ is an extension of shape $(\Sigma\mathbf{SVec})^\times.\pi_0\mathcal{C}$. As in §2.2, we will encode that $\mathcal{C}$ is sylleptic by placing the (invertible) objects $\{\mathbb{1},c\}.\pi_0\mathcal{C}$ in degree 3. In other words, setting $M := \pi_0\mathcal{C}$, we are interested in extensions of shape:

$$(\Sigma\mathbf{SVec})^\times[3].M[3]. \tag{14}$$

The classification of arbitrary extensions of this shape is somewhat complicated. But we know one thing more: the fermion $f$, and hence also its condensate $c$, are invisible. This is sometimes referred to as a local fermion, and any theory with this feature couples to spin structure and is equipped with a $\mathbb{Z}_2$ fermion parity symmetry that induces a grading on the Hilbert space. In the language of group theory, one can think of this as saying that the extension (14) is a "central extension," and so classified by untwisted cohomology (of $M[3]$) with coefficients in $(\Sigma\mathbf{SVec})^\times$.

Cohomology with coefficients in $(\Sigma\mathbf{SVec})^\times$ is called *(extended) supercohomology* $\mathrm{SH}^\bullet$. The name is due to [25] given in the context of condensed matter and lattice constructions, but had appeared in the mathematics literature beforehand as a generalized cohomology theory. See [24] for a more topological treatment. By the Atiyah-Hirzebruch spectral sequence, $\mathrm{SH}^\bullet$ is built out of three "layers" corresponding to the three homotopy groups of $(\Sigma\mathbf{SVec})^\times$. The bottom (Majorana) layer records whether the group of simple objects $\{\mathbb{1},c\}.M$ is or is not a split extension. The second (Gu-Wen) layer records whether the isomorphism given by the braiding on two objects is even or odd; the fermion in particular braids with itself up to a sign rather than braiding trivially. The top layer records the associator data, i.e. a bosonic anomaly, of a suitable bosonic shadow to the fermionic theory [26]. There is a map $\mathrm{H}^\bullet(M[3]; \mathbb{C}^\times) \to \mathrm{SH}^\bullet(M[3])$ corresponding to viewing a bosonic theory as a fermionic one.[9]

**Proposition 2.3.** *For $M$ any arbitrary Abelian group, $\mathrm{SH}^6(M[3]) \cong \widehat{\bigwedge^2 M} = \mathrm{hom}\left(\bigwedge^2 M, \mathbb{C}^\times\right)$, the space of alternating 2-forms.*

*Proof.* We converge to the supercohomology by way of the Atiyah-Hirzebruch spectral sequence $\mathrm{SH}^6(M[3]) \Leftarrow \mathrm{H}^\bullet(M[3]; \mathrm{SH}^\bullet(\mathrm{pt}))$. The entries on the $E_2$ page can be filled in from

---

[9]It was predicted in [27] that the classification of fermionic theories with symmetries in $d$ dimensions is given by (twisted) spin cobordism $\Omega_{\mathrm{Spin}}^{d+1}(M)$. The Atiyah-Hirzebruch spectral sequence then allows us to compile the information in the first three layers to compute an approximation of spin cobordism, this recovers supercohomology. In low dimensions supercohomology well approximates spin cobordism, but as the dimensions get higher, the approximation is more crude and more information coming from the deeper layers may be necessary. In our case however, the supercohomology approximation is exact: while spin cobordism has layers below the Majorana layer, these layers do not contribute to cohomology of $M[3]$ because of the Hurewicz theorem.

the formulas in [20, 21]. This data assembles as:

$$E_2^{i,j} = \begin{array}{c|cccccc} j & & & & & & \\ \hline \mathbb{Z}_2 & \hom(M, \mathbb{Z}_2) & \mathrm{Ext}(M, \mathbb{Z}_2) & \hom(M, \mathbb{Z}_2) & & \dots & \\ \mathbb{Z}_2 & \hom(M, \mathbb{Z}_2) & \mathrm{Ext}(M, \mathbb{Z}_2) & \hom(M, \mathbb{Z}_2) & \widehat{M}_2 \oplus \hom\left(\bigwedge^2 M, \mathbb{Z}_2\right) & & \\ \mathbb{C}^\times & \widehat{M} & 0 & \hom(M, \mathbb{Z}_2) & \widehat{M}_2 \oplus \widehat{\bigwedge^2 M} & & \\ \hline & 3 & 4 & 5 & 6 & i\,. & \end{array} \tag{15}$$

The entries which include hom and Ext in degree three through five are all isomorphic to $\widehat{M}_2$, where $M_2$ denotes the 2-torsion of $M$, and the hat denotes Pontryagin duality. Specifically, $\hom(M, \mathbb{Z}_2) = (\widehat{M})_2$, and $\mathrm{Ext}(M, \mathbb{Z}_2) = \widehat{M}_2$, which can be seen from the short exact sequence $\mathbb{Z}_2 \xrightarrow{(-1)^x} \mathbb{C}^\times \xrightarrow{x^2} \mathbb{C}^\times$. The $d_2$ differential are given by:

$$d_2 : E_2^{i,2} = \mathrm{H}^i(M[3]; \mathbb{Z}_2) \to E_2^{i+2,1} = \mathrm{H}^{i+2}(M[3]; \mathbb{Z}_2) \qquad X \mapsto \mathrm{Sq}^2 X\,, \tag{16}$$

$$d_2 : E_2^{i,1} = \mathrm{H}^i(M[3]; \mathbb{Z}_2) \to E_2^{i+2,0} = \mathrm{H}^{i+2}(M[3]; \mathbb{C}^\times) \qquad X \mapsto (-1)^{\mathrm{Sq}^2 X}\,. \tag{17}$$

Notice that because we are really looking at Eilenberg-MacLane spaces in degree three, we do not need to consider the entries in degree lower than three due to the Hurewicz's theorem:

$$\mathrm{H}^\bullet(M[3]; A) = 0 \quad \text{for } \bullet < 3\,. \tag{18}$$

We claim that $\mathrm{Sq}^2 : \mathrm{H}^3(M[3]; \mathbb{Z}_2) \to \mathrm{H}^5(M[3]; \mathbb{Z}_2)$ is an isomorphism. To see this, note first that $\mathrm{H}^3(M[3]; \mathbb{Z}_2) \cong \hom(M; \mathbb{Z}_2)$ by Hurewicz. Now given $\mu \in \mathrm{H}^3(M[3]; \mathbb{Z}_2)$, we can construct the pullback $\mu^* : \mathrm{H}^\bullet(\mathbb{Z}_2[3]; \mathbb{Z}_2) \to \mathrm{H}^\bullet(M[3]; \mathbb{Z}_2)$. The ring $\mathrm{H}^\bullet(\mathbb{Z}_2[3]; \mathbb{Z}_2)$ is a polynomial ring in the generators $T, \mathrm{Sq}^1 T, \mathrm{Sq}^2 T, \dots$ where $T$ has degree 3. In particular, $\mathrm{H}^3(\mathbb{Z}_2[3]; \mathbb{Z}_2) = \{0, T\}$, with $\mu^*(T) = \mu$, and $\mathrm{H}^5(\mathbb{Z}_2[3]; \mathbb{Z}_2) = \{0, \mathrm{Sq}^2 T\}$. Since $\mathrm{Sq}^2$ is natural, we have $\mathrm{Sq}^2(\mu^* T) = \mu^*(\mathrm{Sq}^2 T)$, confirming the claim. Thus the $d_2$ differentials $E_2^{3,1} \to E_2^{5,0}$ and $E_2^{3,2} \to E_2^{5,1}$ are isomorphisms. The $d_2$ differentials supported in bidegrees $(4, 1)$ and $(4, 2)$ are injections by essentially the same argument. Namely, for each $m \in M_2 = \hom(\mathbb{Z}_2, M)$, we can restrict $\widehat{M}_2 = \mathrm{H}^4(M[3]; \mathbb{Z}_2)$ along the map $m^* : \mathrm{H}^4(\mathbb{Z}_2[3]; \mathbb{Z}_2) \to \mathrm{H}^4(M[3]; \mathbb{Z}_2)$. The only element in $\mathrm{H}^4(\mathbb{Z}_2[3]; \mathbb{Z}_2)$ is $\mathrm{Sq}^1 T$, which is not annihilated by $\mathrm{Sq}^2$. Again by naturality, the $d_2$ from $E_2^{4,1} \to E_2^{6,0}$ and $E_2^{4,2} \to E_2^{6,1}$ are injections.

All together, the $E_3$ page reads:

$$E_3^{i,j} = \begin{array}{c|ccccc} j & & & & & \\ \hline \mathbb{Z}_2 & 0 & 0 & * & * & \\ \mathbb{Z}_2 & 0 & 0 & 0 & * & \\ \mathbb{C}^\times & \widehat{M} & 0 & 0 & \widehat{\bigwedge^2 M} & \\ \hline & 3 & 4 & 5 & 6 & i\,. \end{array} \tag{19}$$

In particular in total degree 6 the spectral sequence stabilizes on page 3, with the only nonzero entry being $\widehat{\bigwedge^2 M}$ in bidegree $(6, 0)$. $\qquad\square$

**Remark.** We note that $\mathrm{H}^6(M[3]; \mathbb{C}^\times) \simeq \widehat{M}_2 \oplus \widehat{\bigwedge^2 M}$ classifies 5d bosonic topological phases, but the $\widehat{M}_2$ is killed by a differential in the spectral sequence for supercohomology. Thus a bosonic sylleptic form contains more information than its superization.

Thus we find:

**Theorem 2.4.** *The set of fermionic (4+1)d topological orders with no lines is equal to the set of symplectic Abelian groups.*

For the definition of *symplectic Abelian group* we refer the reader to §3.1.

*Proof.* The principle of remote detectability for topological orders ensures that there are no invisible operators (trivial centre). In detail, the "trivial centre" requirement for a sylleptic fusion 2-category is that its symmetric centre — its full subcategory on those objects $x$ for which $v_{x||-} = v_{-||x}^{-1}$ — should be trivial. As explained at the end of §2.2, the class in $\widehat{\bigwedge^2 M}$ precisely records the antisymmetric pairing $\langle x, y \rangle = v_{x||y} v_{y||x}$. When applied to the group of surfaces, it means that the symplectic pairing is nondegenerate. $\qquad\square$

**Remark.** To make contact with lower dimensions, consider the familiar case of bosonic 3d topological orders. These are given by modular tensor categories (MTC). In the Abelian case, with $M$ a group, the braiding data of the MTC data is determined by a class in $H^4(M[2])$. This is isomorphic to the group of quadratic functions on $M$. The full braiding of lines is given by the symmetric pairing

$$M \otimes M \to \mathbb{C}^\times, \tag{20}$$
$$a \otimes b \mapsto \frac{q(a+b)}{q(a)q(b)},$$

where $q$ is a quadratic function. In 3d this is a one-parameter family in which the lines pass around each other in a circle and we get a phase factor because it is a one-dimensional motion, and one dimension lower than line operator is a phase. Furthermore, this phase depends symmetrically on the two inputs, and here "nondegenerate" means that the symmetric pairing is nondegenerate.

If $M_2$ is trivial, then $H^4(M[2]; U(1)) \cong \mathrm{Sym}^2\widehat{M}$ by completing the square. In general, the map $H^4(M[2]) \to \mathrm{Sym}^2\widehat{M}$ has kernel. This is analogous to the kernel $\widehat{M}_2$ of the map $H^6(M[2]; U(1)) \to \mathrm{SH}^6(M[3]) = \widehat{\bigwedge^2 M}$. And indeed a similar analysis as in Proposition 2.3 shows that this kernel dies when going to fermionic theories and $\mathrm{SH}^4(M[2]) \cong \mathrm{Sym}^2\widehat{M}$.

# 3 5d Topological order from the boundary

## 3.1 Symplectic Reduction to Isotropic Subspaces

The symplectic form $\omega$ on $M$ gives $M$ the structure of a symplectic Abelian group.

**Definition 3.1.** *A* symplectic Abelian group *is an Abelian group G together with an isomorphism* $\omega : M \to \widehat{M}$, *with* $\widehat{M} = \hom(M, \mathbb{C}^\times)$ *such that* $\omega(g,g) = 1$ *for every* $g \in M$.

This definition implies an alternating feature, $\omega = \widehat{\omega}^{-1}$. Recall that by definition $\bigwedge^2 M = \frac{M \otimes M}{m \otimes m}$, so a map $\omega : M \to \widehat{M}$ is the same data as a map $\omega : M \otimes M \to U(1)$. This map solves $\omega(g,g) = 1$ for all $g$ iff it factors through $\bigwedge^2 M$.

**Example.** An example of a symplectic Abelian group is when $M$ is a product of groups $B \times \widehat{B}$ with $\omega((b_1, f_1), (b_2, f_2)) = f_1(b_2) \cdot f_2(b_1)^{-1}$.

If $M$ is a cyclic group then $M$ does not admit a symplectic form. Call the generator for the cyclic group $t$, then $\omega(t,t) = 1$ but $\omega(t^a, t^b) = 1^{ab} = 1$, and $\omega$ is not an isomorphism.

Suppose M is a symplectic Abelian group and $N \subset M$ is a subgroup. The *symplectic orthogonal* $N^\perp$ is the subgroup $\{m \in M \text{ s.t. } \omega(m, n) = 1 \text{ for all } n \in N\}$. It is the subgroup corresponding to $\widehat{M/N} \subset \widehat{M}$ under the isomorphism $\omega : M \cong \widehat{M}$. From this description, we see that $|M| = |N| \times |N|^\perp$. A subgroup $L \subset M$ is *Lagrangian* if $L = L^\perp$ as subgroups of $M$. Thus $L$ is Lagrangian exactly when $\omega|_L$ is trivial and $|L| = \sqrt{|M|}$. A *Lagrangian splitting* of $M$ is a direct sum decomposition $M = L \oplus L'$ where both $L$ and $L'$ are Lagrangian. The symplectic form on $M$ then identifies $L' \cong \widehat{L}$.

**Proposition 3.2** (Darboux theorem for finite groups)**.** *Every symplectic finite Abelian group admits a Lagrangian splitting.*[10]

The following proof is essentially given in [28, Lemma 5.2].

*Proof.* Every finite Abelian group canonically factors as a direct sum of subgroups for different $p$, and the symplectic form cannot mix different primes. We thus reduce to the case where the group in consideration $M$ has order $p^k$ for some prime $p$. We give the $p = 2$ case for clarity, and the proof generalizes for other primes. Pick an element $x \in M$ of maximal order, say $2^a$. Then $x^{2^{a-1}}$ is nontrivial and we choose an element $y$ such that the pair $\omega(x^{2^{a-1}}, y) \neq 1$. We use the fact that $x^{2^{a-1}}$ is order 2 and so by inspecting $\omega(x^{2^{a-1}}, y) = \omega(x, y)^{2^{a-1}}$, which is itself also order 2, we see that $y$ has order at least $2^a$. But $a$ was maximal, so we have found two subgroups, generated by $x$ and $y$, both of order $2^a$. We note that these two groups are transverse because an alternating form vanishes on a cyclic subgroup. Let $N$ denote the subgroup generated by $x$ and $y$. It is a product of cyclic groups $\mathbb{Z}_{2^a} \times \mathbb{Z}_{2^a}$, which are themselves each Lagrangians in $N$. The restriction of $\omega$ to $N$ is the canonical split pairing $\omega(x, y)$, of $x$ pairing with $y$. By construction, $\omega|_N$ is nondegenerate. Thus $N$ and $N^\perp$ are transverse ($N \cap N^\perp = 0$), so $M = N \oplus N^\perp$. By induction of the previous procedure, $N^\perp$ can be further split into something Lagrangian, therefore $M$ has a Lagrangian splitting. $\square$

## 3.2 Lagrangian subgroups as boundary theories

We now turn to investigate the boundary (3+1)d theory of a 5d theory which also has only surfaces, and make some relations with the bulk. The boundary is a braided strongly fusion 2-supercategory $\mathcal{L}$ with objects being surfaces that have an $L$ group fusion rule. The braiding $\beta$ is a class in $\mathrm{SH}^5(L[2])$, which in this case, antisymmetrically pairs objects.[11] We can think of the bulk (4+1)d theory as the *sylleptic centre* of $\mathcal{L}$ denoted by $\mathcal{Z}_{(2)}(\mathcal{L})$, where the objects have fusion rule $M$ with a sylleptic structure $\omega$. Since all 1-morphisms are trivial in the strongly fusion case, all the data is encoded in $R, S$ and of particular importance is the class in $\mathrm{SH}^5(L[2])$ encoding the braiding.

**Definition 3.3.** *Let $\mathcal{L}$ be a braided monoidal 2-category. An object in the* sylleptic centre $\mathcal{Z}_{(2)}(\mathcal{L})$ *is a pair $(x, v_{x||-})$. A 1-morphism from $(x, v_{x||-})$ to $(x', v_{x'||-})$ is a one morphism $f : x \to x'$ in $\mathcal{L}$ such that the following diagram commutes for all $y \in \mathcal{L}$:*

---

[10]While it is true that all such $M$ admit a Lagrangian subgroup, it is not the case that any Lagrangian at all fits into the sequence $\widehat{L} \hookrightarrow M \to L$, see [28, Example 5.4]

[11]The analogue of this class for a (1+1)d boundary to a (2+1)d bulk would be a class $\alpha \in \mathrm{SH}^3(L[1])$ that provides associator information regarding the lines in the (1+1)d theory.

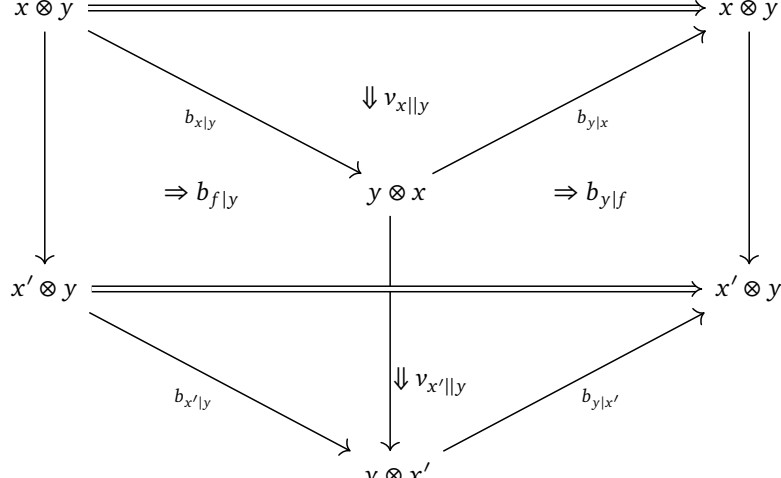

*where the 2-morphism on the back face is the identity. The two morphisms are defined in the same manner as in $\mathcal{L}$.*

**Lemma 3.4.** *If $\mathcal{L}$ is a strongly fusion braided fusion 2-category, then $\mathcal{Z}_{(2)}(\mathcal{L})$ contains no lines.*

*Proof.* Consider the identity object $(\mathbb{1}, v_{\mathbb{1}||-})$ of $\mathcal{Z}_{(2)}(\mathcal{L})$, where $v_{\mathbb{1}||x}$ is the following isomorphism:

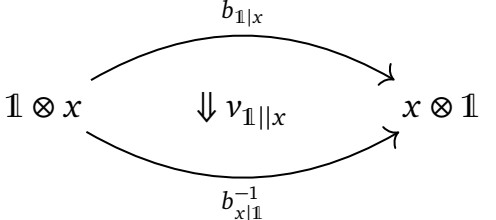

A priori, $v$ could be any $x$-dependent $\mathbb{C}^\times$ number satisfying a 1-cocycle relation i.e. $v \in \widehat{L}$. But since $\mathbb{1}$ is the identity object, $b_{\mathbb{1}|x}$ and $b_{x|\mathbb{1}}$ are both trivialized, and the identity object of the centre is the one such that $v$ is also trivialized, so we take $v_{\mathbb{1}||-} = 1$. Now consider morphisms of the identity object, which is a morphism from $\mathbb{1} \to \mathbb{1}$, or $\mathrm{id}_{\mathbb{1}}$. Then, we have the following 3-cell filling:

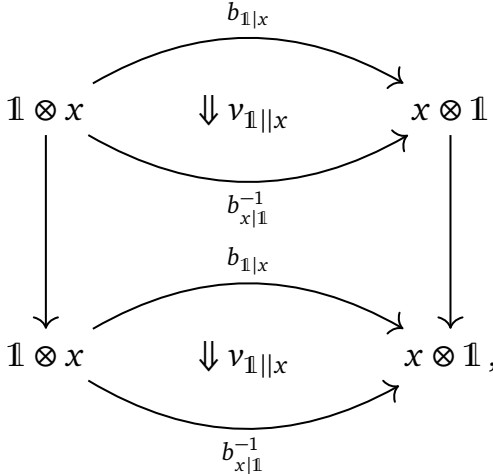

where the vertical maps are just identity maps. But, because we are in the 2-category, the only 3-cell is the identity. $\qquad\square$

**Remark.** More generally, if $\mathcal{B}$ is any braided monoidal 2-category, then $\Omega\mathcal{Z}_{(2)}(\mathcal{B})$ is a full sub-1-category of $\Omega\mathcal{B}$. However, the analogous statements for $\mathcal{Z}_{(1)}$ and for 3-categories fail. For definiteness, Lemma 3.4 is to spell out the details of the case we care about.

**Proposition 3.5.** *The sylleptic center of a trivially braided fusion 2-category $\mathcal{L}$ is $\widehat{\mathcal{L}} \times \mathcal{L}$ and the sylleptic form is the canonical one.*

*Proof.* The trivial braiding indicates that $[\pi, R, S]$ in (11) are trivial. As a consequence, the diagram in (12) reduces down so that $v$ satisfies the equation $v_{\ell||xy} = v_{\ell||x} v_{\ell||y}$, thus $v_{\ell||-}$ is a homomorphism $\mathcal{L} \to \mathbb{C}^\times$. The object $(x, v_{x||-})$ in the sylleptic centre is therefore an element of $(\mathcal{L}, \widehat{\mathcal{L}})$. $\qquad\square$

**Example.** For clarity let us work bosonically in this example instead of using supercategories. Suppose $M$ admits $L$ as a lagrangian, and take $L = \mathbb{Z}_3$. A particularly simple class of braided fusion 2-category is $\mathcal{L} = 2\mathbf{Vec}^\beta[\mathbb{Z}_3]$, where $\beta \in \mathrm{H}^5(\mathbb{Z}_3[2])$. A computation shows that $\mathrm{H}^5(\mathbb{Z}_3[2]) = \widehat{\bigwedge^2 \mathbb{Z}_3} = 0$, which means the only category is $2\mathbf{Vec}[\mathbb{Z}_3]$ with the sylleptic centre

$$\mathcal{Z}_{(2)}(2\mathbf{Vec}[\mathbb{Z}_3]) = \widehat{2\mathbf{Vec}[\mathbb{Z}_3]} \times 2\mathbf{Vec}[\mathbb{Z}_3]. \tag{21}$$

In general, if $L$ was a group such that $\beta \neq 0$, then $\mathcal{Z}_{(2)}(\mathcal{L}) = \widehat{\mathcal{L}}.\mathcal{L}$, a nontrivial extension of the boundary category. In terms of the groups, (21) implies that $M = \widehat{L} \times L$, where $\widehat{L} = M/L^\perp = M/L$. Therefore, $M$ fits into the short exact sequence $\widehat{L} \hookrightarrow M \twoheadrightarrow L$.

**Remark.** The centre gives the corresponding Djikgraaf-Witten (DW) theory for the boundary, with anomaly given by a class in $\mathrm{SH}^6(M[3])$. The act of going from the sylleptic centre to the boundary can be done by first "forgetting" the sylleptic structure, and then applying a Dirichlet boundary condition aka a braided map from a braided monoidal centre to the boundary. The objects in the kernel of this map are precisely the "Wilson lines" of the DW theory. The boundary condition contains not only a condensation $\widehat{L}$ but furthermore a trivialization of $\omega|_{\widehat{L}}$, which is given by a class in $\mathrm{SH}^5(\widehat{L}[3])$.

We can also ask which boundary theories can be lifted to the bulk; this is the equivalent of finding a splitting of the bulk to boundary map. The objects in the image of the splitting map are the "'t Hooft lines" of the DW theory. A priori there can be an obstruction to the lifting [21], which means that the lines in the bulk do not split neatly as a direct sum of "electric" and "magnetic" lines. There exists an obstruction for a braided 4-cocycle $\{\pi(-,-,-,-), R_{(-|-,-)}, S_{(-,-|-)}\}$ to have sylleptic structure given by

$$\theta : \mathrm{H}^5(L[2]) \to \mathrm{Ext}(L, \widehat{L}), \tag{22}$$

with the kernal of this map precisely given by $\widehat{L}_2$. The map $\mathrm{H}^6(L[3]) \to \mathrm{H}^5(L[2])$ maps between the two $\widehat{L}_2$ subgroups, with $\mathrm{H}^6(L[3])$ attained from $\mathrm{H}^6(M[3])$ via a restriction map. The subgroup of $\widehat{L}_2$ in $\mathrm{H}^5(L[2])$ contains information regarding the data of $\pi, R, S$. The remainder of the group is braiding information that cannot be lifted to being sylleptic. There is furthermore a map from $\mathrm{H}^6(L[3]) \to \mathrm{SH}^6(L[3])$ that surjects onto $\widehat{\bigwedge^2 L}$. This is summarized in the following diagram:

$$
\begin{array}{ccc}
\mathrm{H}^6(L[3]) \simeq \underbrace{\widehat{L}_2}_{} \oplus \widehat{\bigwedge^2 L} & \longrightarrow\!\!\!\!\!\rightarrow & \mathrm{SH}^6(L[3]) \simeq \widehat{\bigwedge^2 L} \\
\downarrow & & \downarrow \\
\mathrm{H}^5(L[2]) & \longrightarrow & \mathrm{SH}^5(L[2]).
\end{array}
$$

The map from $\mathrm{SH}^6(L[3]) \to \mathrm{SH}^5(L[2])$ is therefore the zero map as composition of the left vertical map and the horizontal map gives zero; we then have:

**Proposition 3.6.** *Only the fermionic boundary theories with trivial braiding can be extended, in a way such that multiplication data is consistent with the lift to the bulk, to a sylleptic form.*

**Remark.** It is possible that all surfaces on the boundary can be lifted, but not necessarily canonically. In the case of a 3d $\mathbb{Z}_3$ DW theory with nontrivial anomaly, this has a Dirichlet boundary condition where the lines obey $\mathbb{Z}_3$ fusion rule. The bulk however has lines that obey $\mathbb{Z}_9$ fusion rule. Any line on the boundary can be lifted, but there is no way to do this in a way that is compatible with the tensor product. The lines on the boundary cube to the trivial line, but lifting it to the bulk means that the cube is nontrivial.

### 3.3 Morita trivial 5d phases

If $M$ admits $L$ as a Lagrangian subspace then the corresponding 5d topological order upon symplectic reduction is Morita equivalent to the trivial theory. More succinctly this is know as being *Morita trivial*. This reduction procedure is depicted physically in Figure 2.

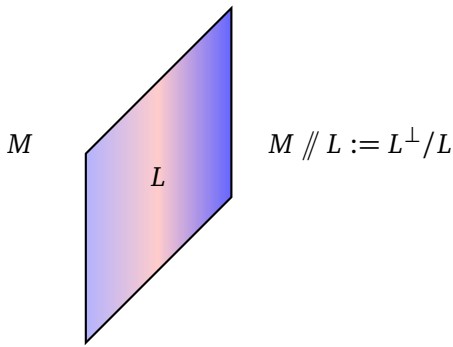

Figure 2: The wall is braided fusion 2-category with objects in $L^\perp$, separating the original theory $\mathcal{A}$ from the vacuum. Similar to the case of quantum Hamiltonian reductions, the wall is a bimodule for the two categories on either side.

**Example.** Consider in (1+1)d the category $\mathcal{I}$ given by $\mathbf{Vec}^{\alpha=0}[\mathbb{Z}_2]$, where $\alpha \in \mathrm{H}^3(\mathbb{Z}_2[1])$ is the trivial associator. Then the (2+1)d bulk theory is $\mathcal{T} = \mathcal{Z}(\mathbf{Vec}[\mathbb{Z}_2]) = \mathbf{Vec}[\mathbb{Z}_2] \times \mathbf{Vec}[\mathbb{Z}_2]$. Condensing out a $\mathbb{Z}_2$ subgroup from $\mathcal{T}$ amounts to the reduction $(\mathbb{Z}_2 \times \mathbb{Z}_2) /\!\!/ \mathbb{Z}_2 = \mathbb{Z}_2^\perp / \mathbb{Z}_2 = \{*\}$. Physically, this is equivalent to taking the (2+1)d Toric code and condensing out the $m$ or $e$ particle. The lines left "unscreened" are in $\mathbb{Z}_2^\perp$, and another identification by $\mathbb{Z}_2$ gives the trivial theory.

Theorem 2.4 relates 5d theories to symplectic Abelian groups and by Proposition 3.2, we see that:

**Proposition 3.7.** *All 5d super topological orders are Morita trivial.*

A 5d phase which is not Morita trivial has boundary conditions that are necessarily gapless, an immediate consequence is:

**Corollary 3.8.** *All 4d fermionic boundaries can be gapped.*

While there are 4d fermionic gapless theories, by introducing the appropriate interactions we can introduce a gap and hence there is no robust gapless phase. [12] We now present the

---

[12] Our definition of topological order is such that an invertible phase is considered as the trivial topological order. In concluding Corollary 3.8 we are using the fact that "topological order" means "topological phase up to invertible phases", and thus for a topological order to have a gapped boundary, this means that the corresponding phase has a gapped interface to an invertible phase.

reverse story and the way of reconstructing a theory from the vacuum. We will show that every fermionic 5d topological order can be built non-canonically by gauging a one-form symmetry $\widehat{L}[1]$, and a zero-form symmetry $G$, both acting on the vacuum. If the set of lines, $\Omega^3 \mathcal{A}$, is super Tannakian, then $G = \mathrm{Aut}(F)$ and $\Omega^3 \mathcal{A} \cong \mathbf{SRep}(G)$. The first step of condensing out the lines can be "undone" by gauging the group $G$ which acts on the group $M$ of surfaces. The symplectic form associated with $M$ is now a $G$-equivariant class $\tilde{\omega} \in \mathrm{H}^6(M[3]/G)$.

After condensing the lines, there is a similar map $C : \Omega^2 \mathcal{A} \to 2\mathbf{SVec}$ which tells how to condense surfaces by choosing Lagrangian subspaces. Gauging by the dual group $\widehat{L}$ which acts on the vacuum then undoes this procedure. For Abelian group, gauging by a group or the dual group is always possible by the notion of "electromagnetic-duality". The important point to stress is that the choice of Lagrangian subgroup $L$ from $M$ was not canonical, and so doing the gauging by $\widehat{L}$ is also not canonical. In contrast, the zero-form group $G$ is canonically determinded based on the lines of $\mathcal{A}$.

**Remark.** This two step procedure can not necessarily be combined to a one step condensation by a "2-group" symmetry $\widehat{L}[1].G$. We take for example the 5d toric code, with a $G = \mathbb{Z}_3$ action that permutes the three strings. A nontrivial extension of $\widehat{L}[1]$ by $G$ will spoil the duality between switching the electic and magnetic lines.

**Example.** We give an analogous story by considering the (2+1)d Toric code. This is only analogous because the theory is not a symplectic Abelian group, rather the pairing is symmetric. We choose the set $\mathcal{R} = \{1, e\}$ to be Tannakian from the set $\{1, e, m, f\}$ of all the lines. As an $\mathcal{R}$ module, the Toric code is $\mathcal{R} \oplus m\mathcal{R}$. The map $F$ takes $R$ and condenses it to the vacuum. This forms a gapped (1+1)d boundary where, as an $\mathcal{R}$ module, the lines $\{1, m\}$ live. The group $G$ is the group generated by $\{1, m\}$, as can be seen when we consider the fact that a zero-form symmetry in (1+1)d is sourced by lines.

**Remark.** Since $M$ is a group of surface operators which are codimension-3 it defines a two-form symmetry. This group has an anomaly that is precisely the symplectic form. For a general isotropic $N \subset M$ we can consider gauging the $N$-symmetry. The importance of being isotropic is to ensure that that the symmetry is non-anomalous and can be gauged. By gauging the symmetry we build a gapped domain wall between the original theory $M$ and a new theory given by $M \sslash N$. In the gauging procedure, $N$ screens out those operators in $M$ which do not commute with $N$, and so the unscreened operators are $N^\perp$. But also the gauging procedure identifies the operators in $N$. The result is that the new procedure is described by the *symplectic reduction* $N^\perp / N$.[13] We note that $M \sslash N$ itself is naturally symplectic by defining $\omega([a], [a']) = \omega(a, a')$ where $[a], [a']$ are classes in $N^\perp / N$, i.e. $a, a' \in N^\perp$, and they are defined up to shifting by $b, b' \in N$. If $N = L$ is Lagrangian, $L^\perp / L = \{*\}$, and so we do not have to assume that $L$ participated in a Lagrangian splitting to show Morita triviality in proposition 3.7.

# 4  Bosonic 5-dimensional Topological Orders

The passage from bosonic to super topological orders is much like going from $\mathbb{R}$ and extending to $\mathbb{C}$, its algebraic closure. Consider a time reversal symmetry $\mathbb{Z}_2^T$ that acts $\mathbb{C}$-antilinearly and squares to the identity. Working with an algebra $A$ of operators over the complex numbers with a $\mathbb{Z}_2^T$ symmetry is the same as working over the real numbers. The $\mathbb{Z}_2^T$ *descends* $A$ into $A_\mathbb{R}$, an $\mathbb{R}$ algebra, so that $A = A_\mathbb{R} \otimes \mathbb{C}$. In the same spirit as the 0-categorical case, there is a way to 1-categorically extend **Vec** to **SVec**, where the latter is "algebraically closed" [29].

---

[13]For an associative algebra $A$, gauging by the action of a connected and simply connected Lie group $G$ is also called *quantum Hamiltonian reduction*.

A bosonic topological order $\mathcal{A}$ is equipped with an action of the categorified Galois group $\mathrm{Gal}(\mathbf{SVec}/\mathbf{Vec}) = \mathbb{Z}_2^F[1]$, and *Galois descent* says that the algebra of a bosonic higher category can be considered as the algebra of a $\mathbb{Z}_2^F[1]$-equivariant higher supercategory. As remarked in Section 2, the fibre functor $F$ may not allow for complete condensation of the lines if we are working bosonically. If the lines are Tannakian i.e. $\mathbf{Rep}(G)$ then we can condense out all the lines, the problem then reduces to the analogous problem discussed in the previous sections, with the symplectic form a class in $\mathrm{H}^6(M[3])$. If the lines are $\mathbf{Rep}(G, z)$, where $z$ is a central element of order two, then it is always possible to condense to only $\{1, f\}$, i.e. $\mathbf{SVec}$. Furthermore, in a 5d bosonic theory not only are there surface operators, but there are nontrivial 3d "membrane" operators. The surfaces operators still form a group under fusion by [18, Theorem A]; in this dimension the surfaces and lines can always unlink. But now either of the lines $\{1, f\}$ can wrap membranes, each detecting the other. This data compiles into a bosonic 3-category $\mathcal{A}$, with $\pi_0\mathcal{A} = \mathbb{Z}_2^F$. The "magnetic membrane" is the unique invertible object in the nontrivial component that enacts the $\mathbb{Z}_2^F$ one-form symmetry and will square to something in the identity component. The whole 3-category is describable by an extension

$$\mathbb{C}^\times[5].\underbrace{\mathbb{Z}_2}_{\{1,f\}}[4].\underbrace{(\mathbb{Z}_2.M)}_{\text{surfaces}}[3].\mathbb{Z}_2^F[2]; \tag{23}$$

$\mathbb{C}^\times[5]$ means "four-form $\mathbb{C}^\times$ symmetry" the $\mathbb{Z}_2$ in surfaces is given by $\{\mathbb{1}, c\}$, which are the two simple objects in $\Sigma\mathbf{SVec}$ as stated before in §2.3, with the caveat that now $c^2 \cong c \oplus c$. The fibre

$$\mathbb{C}^\times[5].\mathbb{Z}_2[4].(\mathbb{Z}_2.M)[3] = (\mathbb{C}^\times[5].\mathbb{Z}_2[4].\mathbb{Z}_2[3]).M[3] \tag{24}$$

is the 2-category of surfaces, and the base $\mathbb{Z}_2^F[2]$ are the two components of the 3-category. We can make a simplification of the fibre as follows. Any surface in $s \in M$ actually corresponds to two surfaces $s_1$ or $s_2$, being off from each other by the $c$. But because we have the magnetic brane, $\mathcal{M}$, we can act with this brane on either of the surfaces. The intersection of $\mathcal{M}$ with $s_1$ or $s_2$ is either the line $1$ or $f$, however we know that $\mathcal{M}$ acting on $c$ gives $f$. Therefore, it is possible to identify which $s_{1,2}$ is the one that is also "charged" with $c$. This gives us the freedom to always choose the "neutral" line, and so the term $\mathbb{Z}_2[3]$ can be ignored. Left with only the surfaces in $M$, we may condense them all out via the procedure in §3.1. We are left with only having to understand the $\mathbb{Z}_2^F[2]$ objects.

The fermionic Witt group inherits an action by $\mathbb{Z}_2^F[1]$ due to the fact that the spectrum [14] $S\mathcal{W}^\bullet = \left(\Sigma^{n-1}\mathbf{SVec}\right)^\times$. $\mathcal{W}^\bullet$ is then the fixed-point spectrum of $\mathbb{Z}_2^F[1]$ via categorified Galois descent [6]. Therefore the cohomology of $\mathcal{W}^\bullet(\mathrm{pt})$ is given by the twisted $S\mathcal{W}^\bullet$-cohomology, $S\mathcal{W}^\bullet(\mathbb{Z}_2^F[2])$, of the space $\mathbb{Z}_2^F[2] = \mathrm{B}(\mathbb{Z}_2^F[2])$. We compute this twisted cohomology by the following Atiyah-Hirzebruch spectral sequence:

$$\mathrm{H}^i(\mathbb{Z}_2^F[2]; S\mathcal{W}^j(\mathrm{pt})) \Rightarrow S\mathcal{W}^{i+j}(\mathbb{Z}_2^F[2]) = \mathcal{W}^{i+j}(\mathrm{pt}). \tag{25}$$

The homotopy groups of $S\mathcal{W}^\bullet(\mathrm{pt})$ in low degrees are given by

$$S\mathcal{W}^0(\mathrm{pt}) = \mathbb{C}^\times, \quad S\mathcal{W}^1(\mathrm{pt}) = \mathbb{Z}_2, \quad S\mathcal{W}^2(\mathrm{pt}) = \mathbb{Z}_2, \tag{26}$$
$$S\mathcal{W}^3(\mathrm{pt}) = 0, \quad S\mathcal{W}^4(\mathrm{pt}) = S\mathcal{W}, \quad S\mathcal{W}^5(\mathrm{pt}) = 0, \quad S\mathcal{W}^6(\mathrm{pt}) = 0.$$

In degree four, $S\mathcal{W}$ known as the *fermionic Witt group* gives the set of (2+1)d super topological orders modulo gapped interfaces. Another way to think about this group is that it gives the

---

[14]Spectrum can be substituted interchangeably with the term "generalized cohomology theory", which was used in the introduction.

anomalies for 3d super MTCs [15], and two theories related by a gapped interface have the same anomaly.

The $E_2$ page is therefore:

$$
E_2^{ij} = \quad
\begin{array}{c|ccccccccc}
j & & & & & & & & & \\
\hline
0 & 0 & 0 & \cdots & & & & & & \\
0 & 0 & 0 & \cdots & & & & & & \\
S\mathcal{W} & S\mathcal{W} & 0 & \mathrm{hom}(\mathbb{Z}_2, S\mathcal{W}) & \cdots & & & & & \\
0 & 0 & 0 & 0 & 0 & 0 & 0 & & & \\
\mathbb{Z}_2 & \mathbb{Z}_2 & 0 & \mathbb{Z}_2 & \mathbb{Z}_2 & \mathbb{Z}_2 & \mathbb{Z}_2^2 & \mathbb{Z}_2^2 \cdots & & \\
\mathbb{Z}_2 & \mathbb{Z}_2 & 0 & \mathbb{Z}_2 & \mathbb{Z}_2 & \mathbb{Z}_2 & \mathbb{Z}_2^2 & \mathbb{Z}_2^2 \cdots & & \\
\mathbb{C}^{\times} & \mathbb{C}^{\times} & 0 & \mathbb{Z}_2 & 0 & \mathbb{Z}_4 & \mathbb{Z}_2 & \mathbb{Z}_2 & \mathbb{Z}_2 & \mathbb{Z}_2 \\
\hline
& 0 & 1 & 2 & 3 & 4 & 5 & 6 & 7 & 8 \quad i\,.
\end{array}
\tag{27}
$$

The *twisted* $d_2$ differentials are:

$$
d_2 : E_2^{i,2} = \mathrm{H}^i(\mathbb{Z}_2^F[2]; \mathbb{Z}_2) \to E_2^{i+2,1} = \mathrm{H}^{i+2}(\mathbb{Z}_2^F[2]; \mathbb{Z}_2) \qquad X \mapsto \mathrm{Sq}^2 X + TX\,, \tag{28}
$$
$$
d_2 : E_2^{i,1} = \mathrm{H}^i(\mathbb{Z}_2^F[2]; \mathbb{Z}_2) \to E_2^{i+2,0} = \mathrm{H}^{i+2}(\mathbb{Z}_2^F[2]; \mathbb{C}^{\times}) \qquad X \mapsto (-1)^{\mathrm{Sq}^2 X + TX}\,,
$$

where $T$ is the generator of $\mathrm{H}^{\bullet}(\mathbb{Z}_2^F[2]; \mathbb{Z}_2)$ in degree two. The $E_3$ page is

$$
E_3^{ij} = \quad
\begin{array}{c|cccccccccc}
j & & & & & & & & & & \\
\hline
0 & 0 & 0 & \cdots & & & & & & & \\
0 & 0 & 0 & \cdots & & & & & & & \\
S\mathcal{W} & S\mathcal{W} & 0 & \mathrm{hom}(\mathbb{Z}_2, S\mathcal{W}) & \cdots & & & & & & \\
0 & 0 & 0 & 0 & 0 & 0 & 0 & & & & \\
\mathbb{Z}_2 & 0 & 0 & \mathbb{Z}_2 & 0 & 0 & \mathbb{Z}_2 & \cdots & & & \\
\mathbb{Z}_2 & 0 & 0 & 0 & \mathbb{Z}_2 & 0 & 0 & 0 & 0 & \cdots & \\
\mathbb{C}^{\times} & \mathbb{C}^{\times} & 0 & 0 & 0 & \mathbb{Z}_4 & \mathbb{Z}_2 & 0 & 0 & 0 & \\
\hline
& 0 & 1 & 2 & 3 & 4 & 5 & 6 & 7 & 8 \quad i\,.
\end{array}
\tag{29}
$$

**Remark.** The generators of $\mathrm{H}^5(\mathbb{Z}_2^F[2]; \mathbb{Z}_2)$ are $\mathrm{Sq}^2 \mathrm{Sq}^1 T$ and $T \mathrm{Sq}^1 T$. The $d_2$ differential annihilates $\left(\mathrm{Sq}^2 \mathrm{Sq}^1 T + T \mathrm{Sq}^1 T\right)$ leaving a $\mathbb{Z}_2$ in bidegree $(5,2)$. The $\mathbb{Z}_2$'s and $\mathbb{Z}_4$ in total degree four survive on $E_{\infty}$ [6, Remark V.2]. The main result in [22] implies that the $\mathbb{Z}_2$ in bidegree $(5, 0)$ survives on $E_{\infty}$.

There is potentially a $d_3$ differential that maps $\mathrm{hom}(\mathbb{Z}_2, S\mathcal{W}) \to E_3^{5,2} = \mathbb{Z}_2$, after which the spectral sequence stabilizes in total degree 6. Thus $\mathcal{W}^6(\mathrm{pt})$ is the kernel of this $d_3$. By [9, Proposition 5.18] we have

$$
S\mathcal{W} = S\mathcal{W}_{\mathrm{pt}} \oplus S\mathcal{W}_2 \oplus S\mathcal{W}_{\infty}\,, \tag{30}
$$

where $S\mathcal{W}_{\mathrm{pt}}$ is generated by the Witt classes of Abelian super MTC, $S\mathcal{W}_2$ is an elementary Abelian 2-group, and $S\mathcal{W}_{\infty}$ is a free group of countable rank. It was proved in [30, Theorem, 7.2] that $S\mathcal{W}_2$ is a group of infinite rank [16], which means that on $E_{\infty}$ the entry in $(2,4)$ will also have infinite rank even after the $d_3$ differential. As a result, $\mathcal{W}^6(\mathrm{pt})$ is also a group of infinite rank. By construction, $\mathcal{W}^6(\mathrm{pt})$ is the group of Morita equivalence classes of 5d topological orders, and so we have verified equation (3):

---

[15]This is a braided fusion category with trivial centre which is equipped with a "ribbon structure," which allows the corresponding (2+1)-dimensional TQFT to be placed on any oriented manifold. The TQFT is said to be *isotropic*.

[16]In particular the spin MTC $\mathrm{SO}(2n + 1)_{2n+1}, n \geq 1$, are pairwise Morita inequivalent.

**Theorem 4.1.** *There are infinitely many 5d bosonic topological orders which are "chiral" in the sense that they only admit gapless boundary.* □

This starkly contrasts our conclusion in section 3.1 for the fermionic case, where $S\mathcal{W}^6(\text{pt})$ was trivial. The source of the difference lies in the fact that the magnetic membrane lives in the bosonic world. If we were to "fermionize" all of the bosonic theories, i.e. couple to spin structure, then the infinite rank group would trivialize.

To gain a more physical intuition for these ungappable and chiral bosonic objects we comment on their construction in a manner similar to [31], used for SPTs. The main takeaway for SPTs is that when constructing an SPT, we can place lower-dimensional invertible phases along homology cycle representatives dual to Stiefel-Whitney classes. This is what was done for the dual of the generalized double semion model in 5d to show that it is equivalent to a twisted Dijkgraaf-Witten dual stacked with lower dimensional SPT phases.

This takeaway leads to a construction of the chiral 5d phases gauranteed by Theorem 4.1. Pick a spin-MTC $\mathcal{C}$ representing an order-2 class in $S\mathcal{W}_2$ that is in the kernel of the $d_3$ differential. We place the 3d topological order built from $\mathcal{C}$ along a representative of $w_2$, by this we mean we place $\mathcal{C}$ along the homology cycle that is dual to $w_2$ (and away from $w_2$ we can just flood the phase with the vacuum). The choice of representative for $w_2$ should not change the theory, the reason for this culminates from the fact that $\mathcal{C}^2$ is super-Witt trivial and furthermore $\mathcal{C}$ is in the kernel of $d_3$. The fact $\mathcal{C}$ is order 2 has to do with protecting our theory under changes of representatives by a $\mathbb{Z}_2^F[1]$-symmetry. Being in the kernel of $d_3$ is telling us that changes of triangulation that might lead to higher order anomalies do not show up.

To see why any 4d boundary theory can not be gapped, note that a representative of $w_2$ in the bulk will end along a representative of $w_2$ on the boundary. But $\mathcal{C}$ is nontrivial, so that representative of $w_2$ on the 4d boundary will necessarily carry a 2d chiral theory, namely a chiral edge mode for $\mathcal{C}$. For instance, suppose $\mathcal{C}$ is $SO(2n+1)_{2n+1}$, or some product thereof that is within the kernel of $d_3$. Then the 4d boundary condition will see chiral WZW modes supported on a representative of $w_2$.

## Acknowledgments

We thank David Reutter for many discussions about fusion 2-categories, and we thank Tian Lan and Diego Delmastro for comments on the draft. Research at the Perimeter Institute is supported by the Government of Canada through Industry Canada and by the Province of Ontario through the Ministry of Economic Development and Innovation. The Perimeter Institute is in the Haldimand Tract, land promised to the Six Nations. Dalhousie University is in Mi'kma'ki, the ancestral and unceded territory of the Mi'kmaq.

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
