# Peer review of "Topological Orders in (4+1)-Dimensions"

_SciPost Physics, doi:SciPost Phys. 13, 068 (2022)_

## Round 1 · Referee Report · Anonymous (Referee 1) · 2022-6-24

Report

This manuscript discussed topological orders in 4+1 dimensions, and calculated the Morita equivalence classes for both super and bosonic cases. It is an important contribution to the understanding of higher dimensional topological orders. The paper is very well organized. The proofs, derivations and calculations all appear sound.

My main concern is about the physical interpretation on the "super" case. Being a trivial n-supercategory nSVec does not physically guarantee gappability. Let's consider the example in 2+1D : if the operators are described by SVec, the corresponding classification (of invertible fermionic topological order) is given by $\mathbb Z$, or $\mathbb Z_{16}$ (modulo fermionic E8 state), and their boundaries are all gapless except the really trivial one with zero chiral central charge. Given such possibility, it is not justified to claim that "All 4d fermionic boundaries can be gapped". Further physical arguments should be provided to support the claim. However, I tend to believe that there are indeed obstructions to gap out the boundary: In analogy to the 16-fold way in 2+1D, the $\mathbb Z_{16}$ could be obtaind as the kernel of $\mathcal W\to \mathcal SW$; the results in this work basically say that in 4+1D $\mathcal SW$ is trivial while $\mathcal W$ is of infinite rank.

Besides, there are quite some typos and minor errors; I list a few below. It is understandable given the complicated technical nature of this work, but I still urge the author to further polish their manuscript. I'm glad to recommend publication after minor revisions.

Requested changes

1- Equations (7)(8) contain many typos. Please double check and correct them. 2- Page 6 near the end of the first paragraph, should it be $\pi_0\mathcal C=M$? 3- Page 7 near the bottom, "The second (Gu-Wen) layer records whether the isomorphism given by the braiding on two objects is even or odd; the fermion in particular braids with itself up to a sign rather than braiding trivially. The top layer records the associator data" I'm confused about this interpretation. My impression is that the first layer is $H^2(M,Z_2)$ for group extension, the second layer is $H^3$ for associator and the third layer is $H^4$ for pentagonator. Why the braiding comes before associator, and pentagonator is missing? Does it have something to do with $M$ in degree 3 instead of 1? 4- Definition 3.3 Should it be "diagram commutes for all $ y \in \mathcal L$"? 5- I'm confused by the proof of Lemma 3.4. It seems that by definition $\Omega \mathcal Z_{(2)}(\mathcal L)$ is a subcategory of $\Omega \mathcal L$, which already proves the Lemma.

  • validity: high
  • significance: high
  • originality: top
  • clarity: good
  • formatting: excellent
  • grammar: perfect

Author:  Matthew Yu  on 2022-07-01  [id 2624]

(in reply to Report 1 on 2022-06-24)
Category:
answer to question
correction

We would like to thank the referee for the comments on our paper. Below we give responses to the points that were made.

  1. Equations 7 and 8 have been revised to correct some minor mistakes in the diagrams

  2. Yes, this has now been fixed.

  3. The referee is correct in saying that the first layer records the group extension; this is the Majorana layer but we call it the "bottom layer", which is a conventional issue of what one decides is the top or bottom. The associator does come after the braiding if we translate to the way the referee wants to describe things. We said the associator was on the top layer, but the referee would say it is in the bottom layer, but this is still consistent since we defined the Majorana to be the bottom, whereas the referee would say that is the top layer.
    As we pointed out at the bottom of page 5, the information regarding the pentagonator and hexagonators group together and recorded at the layer with \bC^\times coefficient, and this is due to the fact that M is in degree 3 instead of 1.

  4. Yes, this has now been fixed.

  5. The referee is correct in making this point, and this is essentially the point of the Lemma. Indeed the k-morphisms in Z_n(L) are a sub of the k-morphisms in L when n=k+1, and are equal when n≥k+2. But it is not a sub when n≤k. We will add a remark before the proof that says: "More generally, if \cB is any braided monoidal 2-category, then \Omega Z_(2)(\cB) is a full sub-1-category of \Omega\cB. However, the analogous statements for Z_(1) and for 3-categories fail. For definiteness, we will spell out the details of the case we care about."

Attachment:

_4_1_d_Topological_Order_Paper.pdf

---

## Round 1 · Referee Report · Anonymous (Referee 2) · 2022-7-12

Report

The paper classifies (up to Morita equivalence) the bosonic and fermionic topological orders in 4+1d. The discussion is rigorous and the results are important, and I recommend publication.

a few questions and minor typos:

  • The classification of bosonic topological orders discussed there contains hat M_2 which is absent in the fermionic counterpart, is it related to the fermionic strings? The fermionic strings are related to w_3=Sq^1w_2 which is trivial in fermionic theories.

  • p12 after (22) "The map from H^6-> H^5 ..." -> "The map H^6->H^5"

-p13 "Z3 many lines", "Z_9 many lines" -> "lines obey Z_3 fusion rule", "lines obey Z_9 fusion rule"

  • p17 the discussion in the last paragraph assume some operator can end on the boundary. But in general, whether a bulk operator can end on the boundary depends on the boundary condition. Perhaps the author can clarify the discussion there.
  • validity: -
  • significance: -
  • originality: -
  • clarity: -
  • formatting: -
  • grammar: -

Author:  Matthew Yu  on 2022-07-19  [id 2670]

(in reply to Report 2 on 2022-07-12)

We thank the referee for the questions and for catching the typos: 1: This \widehat{M}_2 is detected via wrapping a surface on a nonorientable manifold, in particular the Klein bottle. One can think of this as a type of anomaly indicator, where the Klein bottle wrapped with this surface will evaluate to a U(1) number. In this way, we see that this piece captures only braiding data. In the fermionic case, the emergent fermion that was in the bosonic case is actually invisible (i.e. local) so when two surfaces braid and intersect along a line, this lines is not the emergent fermion, which can be thought of as a topological line with spin 1/2, as it could have been in the bosonic case. 

2: This has now been fixed

3: This has now been fixed

4: In this last paragraph, we were first placing a nontrivial 3d theory on the Poincare dual of w_2. Now w_2 in the bulk extends to some w_2 of the boundary, so we can also have some cycle that is dual of this boundary w_2. We are implicitly choosing a Dirichlet like boundary for the nontrivial bulk theory, which gives a 2d WZW like theory.

---

## Editorial Decision

published